# Unveiling Species Diversity Within Early-Diverging Fungi from China VIII: Four New Species in *Mortierellaceae* (*Mortierellomycota*)

**DOI:** 10.3390/microorganisms13061330

**Published:** 2025-06-07

**Authors:** Xin-Yu Ji, Yang Jiang, Fei Li, Zi-Ying Ding, Zhe Meng, Xiao-Yong Liu

**Affiliations:** 1College of Life Sciences, Shandong Normal University, Jinan 250358, China; ji15965902393@163.com (X.-Y.J.); jiangyang202309@126.com (Y.J.); lifeisdnu@126.com (F.L.); 15270343451@163.com (Z.-Y.D.); zmeng@sdnu.edu.cn (Z.M.); 2Institute of Microbiology, Chinese Academy of Sciences, Beijing 100101, China

**Keywords:** *Linnemannia*, *Mortierella*, *Mortierellales*, multi-gene phylogeny, taxonomy

## Abstract

The fungal family *Mortierellaceae* represents ubiquitous and ecologically significant components of soil ecosystems across terrestrial habitats. Through an integrative taxonomic approach combining multi-locus phylogenetic analyses (ITS, LSU, SSU rDNA, *RPB1*, and *Act*) with detailed morphological examinations of rhizosphere soil isolates, four novel species within this family were proposed. This study elucidates the morphological distinctions of novel species from allied species and the phylogenetic relationships among the novel and existing species within the family. *Linnemannia rotunda* sp. nov. (closely related to *L. longigemmata*) is distinguished by its globose sporangia and sporangiospores. *Mortierella acuta* sp. nov. (clustering with *M. yunnanensis*) is characterized by spiky collarettes. *Mortierella oedema* sp. nov. (a sister to *M. macrocystis*) exhibits distinctive ampulliform swellings. *Mortierella tibetensis* sp. nov. (clustering with *M. parvispora*) is named for its geographic origin in Tibet. As the eighth installment in a systematic investigation of early diverging fungal groups in China, this work expands the global taxonomic inventory of *Mortierellaceae* to 148 species, underscoring the ongoing discovery of cryptic biodiversity within this ecologically pivotal group.

## 1. Introduction

The members of the *Mortierellaceae* family are ubiquitous and widely distributed [1,2]. GBIF database documents *Mortierellaceae* from Estonia (92,697 records), Australia (35,514), Czechia (27,899), Russian Federation (15,180), Colombia (13,733), United States of America (13,424), Italy (13,417), Lithuania (11,343), Sweden (11,074), and Norway (422) (https://www.gbif.org/, accessed on 24 February 2025). Because of the ability to grow at a wide range of temperatures (one of the reasons), they can occupy a wide range of habitats. *Mortierellaceae* species are considered to be important saprotrophs [3], usually detected and isolated from soil, plant remains, insect guts, mosses, and living plant roots [4,5,6]. Recent studies on soil microbial communities across the globe have shown that species of *Mortierellaceae* are important members of the soil core microbiome [7,8]. They live in the winter-active soil microbial community, forming substantial fungal biomass in the soil during both the snow-covered and the vegetative periods [9]. They produce polyunsaturated fatty acids, such as arachidonic acid, which are crucial for several biological functions in mammals [10,11,12]. These biological functions are widely used in commercial production, for example, in the production of biofuels [13,14]. Many *Mortierellaceae* species have the ability to promote plant growth, to decompose plant litter, and to remodel rhizosphere microbial communities [15,16]. Some species are also biological control agents, producing active antimicrobial metabolites [17]. In culture, species from the family *Mortierellaceae* typically form white, cottony, zonate or rosette-like colonies, with a distinctive odor reminiscent of garlic or a freshly bathed dog [3,4,5].

Over the past few years, *Mortierellaceae* has experienced an influx of a large number of new species [18]. It currently accommodates 17 genera and 144 species. Among them, the *Mortierella* is the most species-rich genus, with 80 species. It is followed by *Linnemannia*, with 24 species recorded (https://www.catalogueoflife.org/, accessed on 15 February 2025).

In this paper, four new species, *Linnemannia rotunda* sp. nov., *Mortierella acuta* sp. nov., *M. oedema* sp. nov., and *M. tibetensis* sp. nov., were described from soil samples in China (Yunnan, Shandong, and Tibet) based on evidence of molecular phylogeny, morphological characteristics, and growth conditions. The purpose of this study is to integrate new resources of *Mortierellaceae* in China based on polygenic phylogeny and species morphology. This is the eighth report of a serial work on the diversity of Chinese early-diverging fungi [19,20,21,22,23,24,25].

## 2. Materials and Methods

### 2.1. Isolation and Morphology

#### 2.1.1. Isolation

In 2024, soil samples were collected in Yunnan, Tibet, and Shandong, following the methods by Zou et al. [26] and Liu et al. [27]. Each sample (approximately 100 g) was placed into a sterile bag and labeled with date, vegetation type, altitude, latitude, and longitude. All samples were stored at 4 °C after being transported to the laboratory. The strains were isolated from the soil samples using a combination of soil dilution plating and moist-chamber cultivation methods [28]. Approximately 1 g of soil sample was placed into a 10 mL centrifuge tube containing 10 mL of sterile water and agitated on a shaker for 25 min to prepare a soil suspension. One milliliter of the initial suspension was added to nine milliliters of sterile water to obtain a 10^−2^ soil suspension. The process was repeated to achieve 10^−3^ and 10^−4^ soil suspensions. Approximately 200 μL of the 10^−3^ and 10^−4^ soil suspensions were placed in rose bengal chloramphenicol agar (RBC: peptone 5.00 g/L, KH_2_PO_4_ 1.00 g/L, MgSO_4_·7H_2_O 0.50 g/L, rose bengal 0.05 g/L, glucose 10.00 g/L, chloramphenicol 0.10 g/L, agar 15.00 g/L), and evenly dispersed using a sterile triangular glass spreader. [29] The plates were cultivated at 16 °C in the dark for 2–5 d. Subsequently, the agar containing mycelia at the edge of the colonies was transferred to fresh potato dextrose agar (PDA: glucose 20 g/L, potato 200 g/L, agar 20 g/L). Macroscopic images were captured using a digital camera (Canon PowerShot G7X, Canon, Tokyo, Japan). For the moist-chamber method, 1 g of soil was evenly spread on the surface of PDA plates, sealed with parafilm, and incubated invertedly at 16 °C in the dark. After 2–3 d, target strains were purified by streaking with an inoculation loop. Two days later, the agar containing mycelia at the colony edge was transferred to fresh PDA and cultured as described above. Concerning the colony descriptions at 16 °C—in the «Compendium of Soil Fungi», *Mortierella* is mostly cultured at 15–20 °C, where the spore structure is easier to observe. The temperature of 16 °C is not the “optimal growth temperature” for each new species, but it is more stable when observing the morphological characteristics of colonies at 16 °C (e.g., sporangia color, hyphal separation state), which is more conducive to microscopic observation and taxonomic identification.

#### 2.1.2. Morphology

A drop of lactic acid, phenol, and cotton blue staining solution was mounted on the glass slide. Then, a small piece of tape was touched to the surface of the mycelia, making part of the hyphae adhere to it. It was then soaked in the lactic acid, phenol, and cotton blue staining solution. The microscopic morphological characteristics of the fungi were observed using a stereoscope (Olympus SZX10, OLYMPUS, Tokyo, Japan) and a light microscope (Olympus BX53, OLYMPUS, Tokyo, Japan), and images were captured with a high-definition color digital camera (Olympus DP80 OLYMPU, Tokyo, Japan) [20,21,22,23,24,25,30]. Structural measurements were conducted using Digimizer software (v5.6.0), with at least 25 individuals measured for each trait. The minimum and maximum growth temperatures were determined using a gradient method. The culture was initially incubated at 10 °C for two days, and then the temperature was reduced by 1 °C each day until there was no further growth. This temperature was defined as the minimum growth temperature. The culture was initially incubated at 25 °C for two days, and then the temperature was increased by 1 °C each day until there was no further growth. This temperature was defined as the maximum growth temperature. All strains were kept in the preservation tubes containing 10% glycerin at −20 °C and preserved in the Shandong Normal University Culture Collection (XG). The living cultures were stored in the China Microbiological Culture Collection Center, Beijing, China (CGMCC). Dried specimens of strains were submitted to the Herbarium Mycologicum Academiae Sinicae, Beijing, China (Fungarium; HMAS). The taxonomic information was deposited in the Fungal Names repository (https://nmdc.cn/fungalnames/, accessed on 24 February 2025).

### 2.2. DNA Extraction, PCR Amplification, and Sequencing

The DNA extraction kit (Cat. No. 70409-20; BEAVER Biomedical Engineering Co., Ltd., Suzhou, China) was employed for genomic DNA extraction, following the manufacturer’s instructions. The ITS, LSU, SSU, *RPB1*, and *Act* regions were amplified using the primer pairs, with the programs specified in Table 1. The final volume of the PCR reaction mixture was 25 μL, comprising 12.5 μL of 2 × Hieff Canace Plus PCR Master Mix with dye (Yeasen Biotechnology, Shanghai, China, Cat. No. 10154ES03), 9.5 μL of ddH_2_O, 1 μL of forward primer (10 μM; TsingKe, Beijing, China), 1 μL of reverse primer (10 μM), and 1 µL of template genomic DNA (1 ng/μL). DNA fragments were stained with TS-GelRed Nucleic Acid Gel Stain (10,000× in water; TSJ002; Beijing Tsingke Biotech Co., Ltd., Beijing, China). PCR products were visualized at 254 nm on 2% agarose electrophoresis gel [31] and purified using a gel extraction kit (Cat# AE0101-C, Shandong Sparkjade Biotechnology Co., Ltd., Jinan, China). DNA sequencing was performed by Tsingke Biotechnology (Beijing, China). All sequences generated in this study were deposited in GenBank.

### 2.3. Phylogenetic Analyses

Newly acquired sequence data were processed using MEGA v.7.0 to ensure consistency [36,37]. Reference sequences were downloaded from GenBank according to a study on *Mortierellaceae* by Telagathoti et al. [18]. Phylogenetic analyses were conducted for each marker, as well as a concatenation of ITS-LSU-SSU-*RPB1*-*Act*. The phylogeny of *Mortierellaceae* was inferred using both maximum likelihood (ML) and Bayesian inference (BI) algorithms [38,39]. These algorithms were integrated with the CIPRES Science Portal (https://www.phylo.org/, accessed 15 February 2025). ML analysis was carried out with 1000 bootstrap replicates using RaxML 8.2.4 (https://www.phylo.org/) in CIPRES Science Gateway V. 3.3 [40,41]. BI analysis was performed using the GTR + I + G model and sampling frequency of once per 1000 generations. Eight cold Markov chains were run simultaneously for two million generations [42,43]. The resulting phylogenetic trees were optimized with iTOL (https://itol.embl.de, accessed 15 February 2024) and refined using Adobe Illustrator CC 2019 [20].

## 3. Results

### 3.1. Molecular Phylogeny

For *Linnemannia*, phylogenetic analyses were performed on a dataset containing 31 strains, representing 25 species, with *Mortierella cogitans* (CBS 879.97) as an outgroup. The sequence matrix comprises a total of 4527 concatenated characters: 1–634 (ITS), 635–1622 (LSU), 1623–2498 (SSU), 2499–3700 (*RPB1*), and 3701–4527 (*Act*). Among these characters, 829 are parsimony informative, along with 3530 constant and 168 parsimony uninformative. Bayesian tree topology is congruent with that of the ML tree (Figure 1).

For *Mortierella*, phylogenetic analyses were performed on a dataset containing 87 strains, representing 74 species, with Umbelopsis isabelline (NRLL 1757) and U. actotrophica (CBS 31093) employed as outgroups. The sequence matrix comprises a total of 4527 concatenated characters: 1–990 (ITS), 991–1959 (LSU), 1960–3042 (SSU), 3043–4401 (*RPB1*), and 4402–5273 (*Act*). Among these, 1828 are parsimony informative, along with 2542 constant and 903 parsimony uninformative. Bayesian tree topology is consistent with the ML tree (Figure 2).

### 3.2. Taxonomy

#### 3.2.1. *Linnemannia rotunda* X.Y. Ji, Y. Jiang & X.Y. Liu, sp. nov.

Fungal Names—FN 572402

Type—China, Yunnan Province, Yuxi City, Xinping Dai Autonomous Country (23°56′39″ N, 101°30′1″ E, altitude 2397.53 m), from soil, 14 May 2024, X.Y. Ji, holotype HMAS 353518, ex-holotype living culture CGMCC 3.28764 (=XG08755-7-1). Soil sampling was conducted in the humus layer (0–20 cm) in a secondary broad-leaved forest, characterized by mor-type humus. According to the World Reference Base for Soil Resources (WRB), the soil is classified as Acrisols, with diagnostic horizons including an argic horizon and low-activity clays. The pH of the soil ranges from 4.5 to 5.5.

Etymology—the “*rotunda*” (Lat.) refers to the round shape of sporangia and sporangiospores.

Description—colonies on PDA at 16 °C for 5 d, reaching 88 mm diameter, fast growing with a rate of 17.6 mm/d, garlic smell, with sparse aerial mycelia. Hyphae hyaline, 1.7–9.4 µm in diameter, sometimes swollen. Sporangiophores are erect or slightly bent, unbranched, 25.5–146.0 µm long, 1.9–5.0 µm wide, sometimes with a swelling beneath sporangia. Sporangia oval to round, smooth, multi-spored, 10.4–22.3 µm long, 10.7–22.6 µm wide. Columellae present but usually tiny. Sporangiospores smooth, hyaline, mostly round, 9.6–19.0 µm in diameter. Chlamydospores present, mostly oval, 10.1–22.0 µm long, 6.6–15.6 µm wide. Zygospores not found (Figure 3).

Temperature requirements—minimum and maximum growth temperatures were 4 °C and 28 °C, respectively.

Additional specimen examined—China, Yunnan Province, Yuxi City, Xinping Dai Autonomous Country (23°56′39″ N, 101°30′1″ E, altitude 2397.53 m), from soil, 14 May 2024, X.Y. Ji and X.Y. Liu, living culture XG08755-7-2.

Notes—the ITS rDNA phylogenetic analysis showed that the new species *Linnemannia rotunda* is closely related to *L. longigemmata* (MLBV = 78, BIPP = 0.96, Figure 1). The new species is distinguished from *L. longigemmata* by 61/634 characters. Morphologically, compared to *L. longigemmata*, the new species has shorter sporangiophores (25.5–146.0 µm vs. 50–150.0 µm).

#### 3.2.2. *Mortierella acuta* X.Y. Ji, Y. Jiang, and X.Y. Liu, sp. nov.

Fungal Names—FN 572400

Type—China, Shandong Province, Tai’an City, Mount Tai (36°11′49″ N, 117°7′16″ E, altitude 155.8 m), from soil, 12 March 2024, X.Y. Ji, holotype HMAS 353516, ex-holotype living culture CGMCC 3.28761 (=XG08182-4-1). Soil sampling was conducted in the litter layer (5–15 cm) in a coniferous-broadleaf mixed forest, characterized by mull-type humus. According to the World Reference Base for Soil Resources (WRB), the soil is classified as Luvisols, with diagnostic features including a cambic horizon, calcic properties, and clay translocation. The pH of the soil ranges from 6.8 to 7.4.

Etymology—the epithet “*acuta*” (Lat.) refers to the spiky collarette.

Description—colonies on PDA at 16 °C for 6 d, attaining 64 mm diameter, moderately fast growing with a rate of 10.6 mm/d, garlic smell, white cottony, with a rosette pattern, luxuriant and velvety after 20 d of cultivation. Hyphae hyaline, upright or bent. Sporangiophores arising from aerial mycelia, erect or slightly bent, unbranched, 22.4–72.4 µm in height, tapering from 2.8 to 4.3 µm at the base and 1.0 to 1.4 µm at the apex. Sporangia are almost spherical in shape, smooth, deliquescent, multi-spored, and 9.5–12.6 µm in diameter. Columellae were absent. Collarettes were present. Sporangiospores were transparent, mostly oval, 2.6–3.4 µm long, and 1.4–1.7 µm wide. Chlamydospores were present, 6.1–8.2 µm long, and 4.8–6.6 µm wide. Zygospores were absent (Figure 4).

Temperature requirements—minimum and maximum growth temperatures 4 °C and 29 °C, respectively.

Additional specimen examined—China, Shandong Province, Tai’an City, Mount Tai (36°11′49″ N, 117°7′16″ E, altitude 155.8 m), from soil, 12 March 2024, X.Y. Ji and X.Y. Liu, living culture XG08182-4-2.

Notes—phylogenetic analysis of the two combined genes of ITS and LSU showed that the new species *M. acuta* forms an independent and fully supported clade (MLBV = 100, BIPP = 1.00). The new species is distinguished from *M. yunnanensis* by 35/609 and 38/1002 characters in ITS and LSU sequences, respectively. Morphologically, compared to *M. yunnanensis*, the new species produces sporangiospores within 2 weeks, whereas *M. yunnanensis* is cultured for 12 weeks without producing sporangiospores under the same culture conditions (PDA medium at 16 °C, dark incubation). *M. acuta* also produces chlamydospores.

#### 3.2.3. *Mortierella oedema* X.Y. Ji, Y. Jiang, and X.Y. Liu, sp. nov.

Fungal Names—FN 572401

Type—China, Tibet, Shigatse City, Yadong County (27°24′37″ N, 88°54′23″ E, altitude 3535 m), from soil, 25 June 2024, X.Y. Ji, holotype HMAS 353517, ex-holotype living culture CGMCC 3.28762 (=XG00420-1-1). Soil sampling was conducted in the alpine meadow-shrub transition zone (0–10 cm), characterized by moder-type humus. According to the World Reference Base for Soil Resources (WRB), the soil is classified as Cryosols. The pH of the soil ranges from 6.0 to 6.8.

Etymology—the “*oedema*” (Lat.) refers to the swelling of hyphae.

Description—colonies on PDA at 16 °C for 5 d, reaching 45 mm diameter, slow growing with a rate of 9 mm/d, garlic smell, sparse aerial mycelia, with characteristic rosette pattern. Hyphae were hyaline, light brown with age, 1.7–5.1 µm wide, sometimes swollen. Sporangia were oval or spherical, smooth, deliquescent, hyaline, and 30.7–42.5 µm in diameter. Columellae were absent. Collarettes were absent. Sporangiospores were hyaline, smooth, oval, or round, 2.2–3.1 µm long, 2.0–2.9 µm wide. Chlamydospores were absent. Zygospores were not found (Figure 5).

Temperature requirements—minimum and maximum growth temperatures were 4 °C and 28 °C, respectively.

Additional specimen examined—China, Tibet, Shigatse City, Yadong County (27°24′37″ N, 88°54′23″ E, altitude 3535 m), from soil, 25 June 2024, X.Y. Ji, living culture XG00420-1-2.

Notes—the ITS rDNA phylogenetic analysis showed that the new species M. oedema is closely related to *M. macrocystis* (MLBV = 100, BIPP = 1, Figure 2). It is distinguished from *M. macrocystis* by 48/631 characters in ITS sequences. Sporangiospores of the new species are oval or round, while sporangiospores of *M. macrocystis* are almost spherical. 

#### 3.2.4. *Mortierella tibetensis* X.Y. Ji, Y. Jiang, and X.Y. Liu, sp. nov.

Fungal Names—FN 572399

Type—China, Tibet, Shigatse City, Yadong County, (27°21′53″ N, 88°58′26″ E, altitude 3535 m), from soil, 26 June 2024, X.Y. Ji, holotype HMAS 353519, ex-holotype living culture CGMCC 3.28763 (=XG00421-2-1). The soil sampling sites and characteristics are the same as those of *M. oedema*.

Etymology—the “*tibetensis*” (Lat.) refers to the Tibet Autonomous Region of China, where the type was collected.

Description—colonies on PDA at 16 °C for 7 d, reaching 59 mm diameter, slow growing with a rate of 8.4 mm/d, garlic smell and a wet dog smell, with sparse aerial mycelia. Hyphae were hyaline, upright or bent, and sometimes swollen. Sporangiophores arose from aerial mycelia and were erect or slightly bent, unbranched, 112–406 µm in height, and had tapering from 4.2 to 6.9 µm at the base and 1.6 to 2.6 µm at the apex. Sporangia were almost spherical in shape, smooth, deliquescent, multi-spored, and 12.8–30.3 µm in diameter. Columellae were absent. Collarettes were absent. Sporangiospores were hyaline, mostly pentagonal or hexagonal, smooth, and 2–5.9 µm long. Chlamydospores were absent. Zygospores were not observed (Figure 6).

Temperature requirements—minimum and maximum growth temperatures of 4 °C and 29 °C, respectively.

Additional specimen examined—China, Tibet, Shigatse City, Yadong County (27°21′53″ N, 88°58′26″ E, altitude 3535 m), from soil, 26 June 2024, X.Y. Ji and X.Y. Liu, living culture XG00421-2-2.

Notes—phylogenetic analysis of three loci (ITS, LSU, SSU) showed that the new species M. tibetensis is closely related to *M. parvispora* (MLBV = 88, BIPP = 1, Figure 2). The new species is distinguished from *M. parvispora* by 49/648, 18/969, and 25/1019 characters in ITS, LSU, and SSU sequences, respectively. The new species has larger pentagonal or hexagonal sporangiophores, while sporangiospores of *M. parvispora* are smaller (2–3 μm) and almost spherical and subspherical.

## 4. Discussion

With the development of molecular biology techniques, the taxonomy of the family *Mortierellaceae* has gradually shifted from traditional morphology-based classification to a comprehensive classification system that integrates both morphology and molecular data [6,44]. This study has shown that, based on the multigene phylogenetic analysis, the genera *Dissophora*, *Gamsiella*, and *Lobosporangium* have been merged with the genus *Mortierella*, resulting in the genus *Mortierella* no longer being a monophyletic group [5]. This phylogenetic analysis provides a more accurate framework for the classification of the family *Mortierellaceae*. The four new species of the family *Mortierellaceae* described in the study also used both morphological and molecular characteristics for identification and differentiation. In addition to the traditional ITS locus, other loci (LSU, SSU rDNA, *RPB1,* and *Act*) were employed to construct the phylogenetic trees, which helps to more accurately identify and distinguish the new species within the family *Mortierellaceae*. In recent years, the phylogenetic analysis of the *Mortierellaceae* family has primarily relied on morphological characteristics and the ITS + LSU sequences [45]. Basically, the results were consistent with previous studies based on ITS + LSU.

Globally, new species of the family *Mortierellaceae* are constantly being discovered and described. For example, in China, several new recorded species and putative new species of the genus *Mortierella* were described [3,4,5,18]. These new findings enrich our understanding of the biodiversity of the *Mortierellaceae* family. They also indicate that this family has a wide distribution and a complex evolutionary history in different ecosystems. Species of the *Mortierellaceae* family exhibit strong adaptability to different environments. For example, some species are active in alpine and polar environments, while others are widespread in temperate and tropical soils [46,47,48,49]. The four new species described in this study were isolated from the soil samples in Yunnan Province, Tibet Autonomous Region, and Shandong Province. This finding enriches our understanding of the diversity, distribution, and ecological preferences of the *Mortierellaceae*.

## Figures and Tables

**Figure 1 microorganisms-13-01330-f001:**
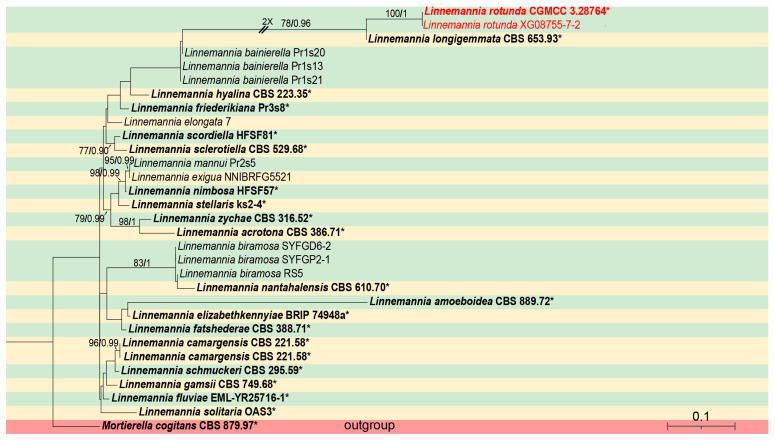
The ML phylogenetic tree of the genus *Linnemannia* based on the concatenated alignment of ITS, LSU, SSU, *RPB1,* and *Act* sequences, with *Mortierella cogitans* serving as outgroup. Branches are labeled with Maximum Likelihood Bootstrap Value (left, MLBV ≥ 70) and Bayesian Inference Posterior Probability (right, BIPP ≥ 0.90), which are separated by a slash “/”. New species are highlighted in red. Branches shortened due to space constraints are indicated by double slashes “//” and the number of folds. Strains marked with an asterisk “*” and in bold represent ex-type or ex-holotypes. The bottom-right scale bar indicates 0.1 substitutions per site.

**Figure 2 microorganisms-13-01330-f002:**
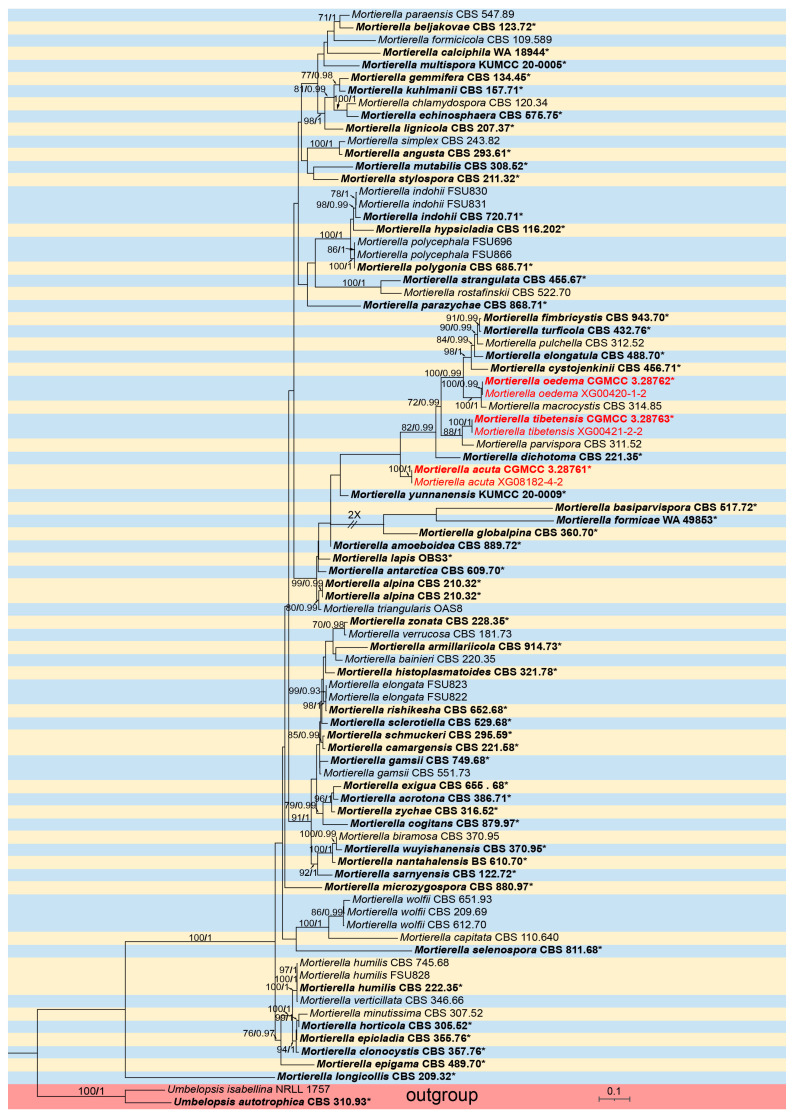
The ML phylogenetic tree of the genus *Mortierella* based on the concatenated alignment of ITS, LSU, SSU, *RPB1*, and *Act* sequences, with *Umbelopsis isabellina* and *U. actotrophica* serving as outgroups. Branches are labeled with Maximum Likelihood Bootstrap Value (left, MLBV ≥ 70) and Bayesian Inference Posterior Probability (right, BIPP ≥ 0.90), which are separated by a slash “/”. New species are highlighted in red. Branches shortened due to space constraints are indicated by double slashes “//” and the number of folds. Strains marked with an asterisk “*” and in bold represent ex-types or ex-holotypes. The bottom-right scale bar indicates 0.1 substitutions per site.

**Figure 3 microorganisms-13-01330-f003:**
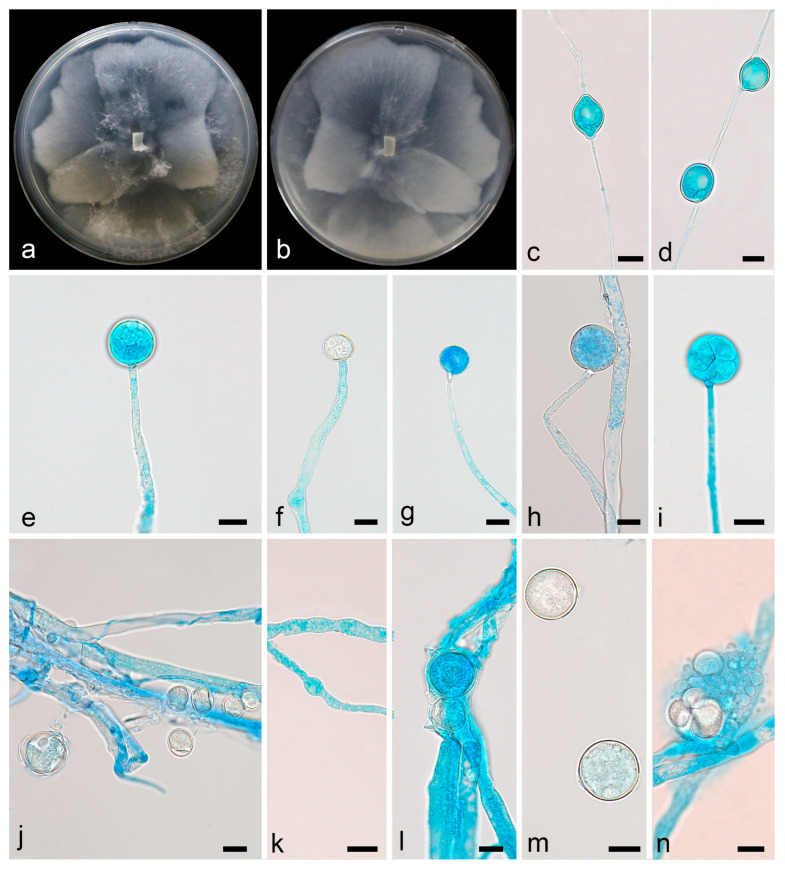
*Linnemannia rotunda* ex-holotype CGMCC 3.28764. (**a**,**b**) Colonies on PDA (**a** obverse, **b** reverse); (**c**,**d**) chlamydospores; (**e**–**i**) sporangia; (**k**) typical swollen hyphae; (**j**,**l**–**n**) sporangiospores; scale bars: (**c**–**n**) 10 µm.

**Figure 4 microorganisms-13-01330-f004:**
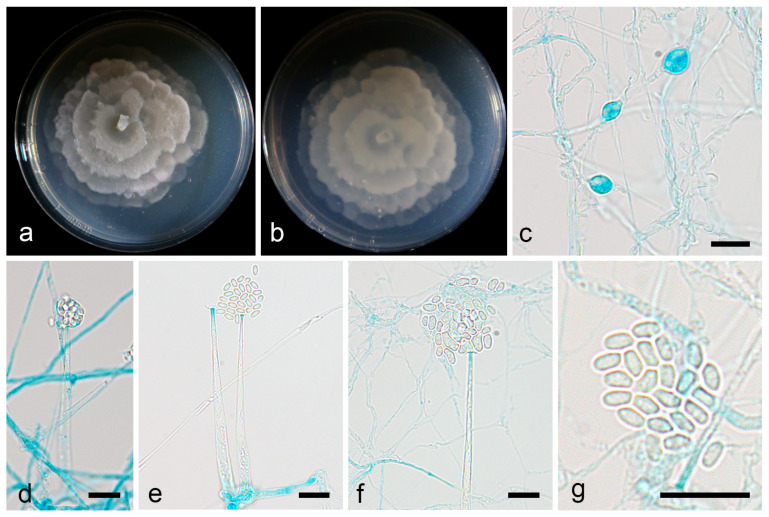
*Mortierella acuta* ex-holotype CGMCC 3.28761. (**a**,**b**) Colonies on PDA (**a** obverse, **b** reverse); (**c**) chlamydospores; (**d**) sporangia; (**e**,**f**) deliquescent sporangia releasing sporangiospores and leaving obvious collarettes; (**g**) sporangiospores; scale bars: (**c**–**g**) 10 µm.

**Figure 5 microorganisms-13-01330-f005:**
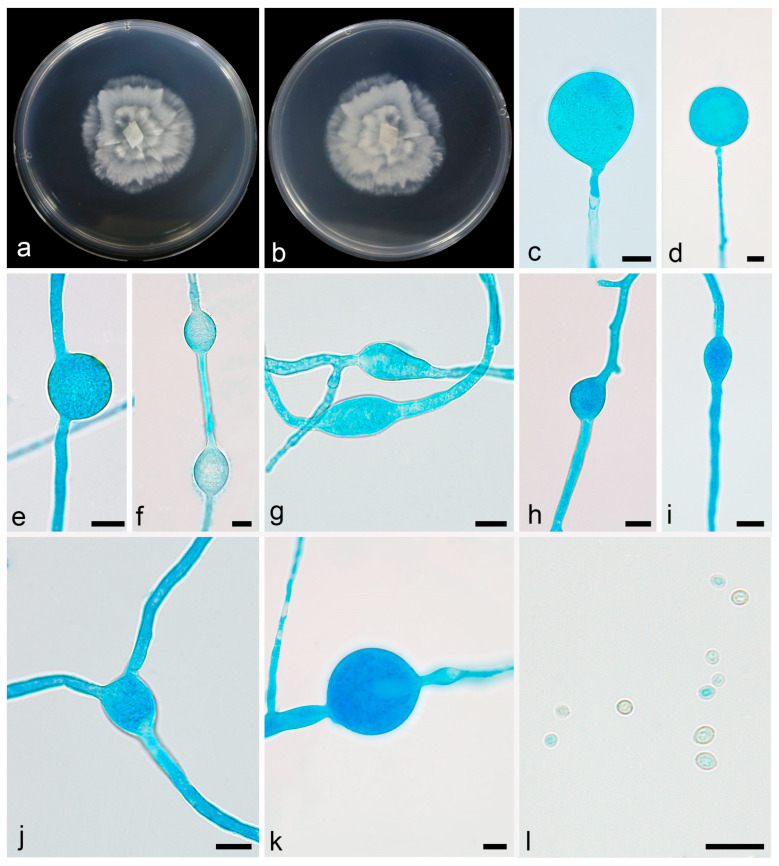
*Mortierella oedema* ex-holotype CGMCC 3.28762. (**a**,**b**) Colonies on PDA (**a** obverse, **b** reverse); (**c**,**d**) sporangia; (**e**–**k**) typical swollen hyphae; (**l**) sporangiospores; scale bars: (**c**–**l**) 10 µm.

**Figure 6 microorganisms-13-01330-f006:**
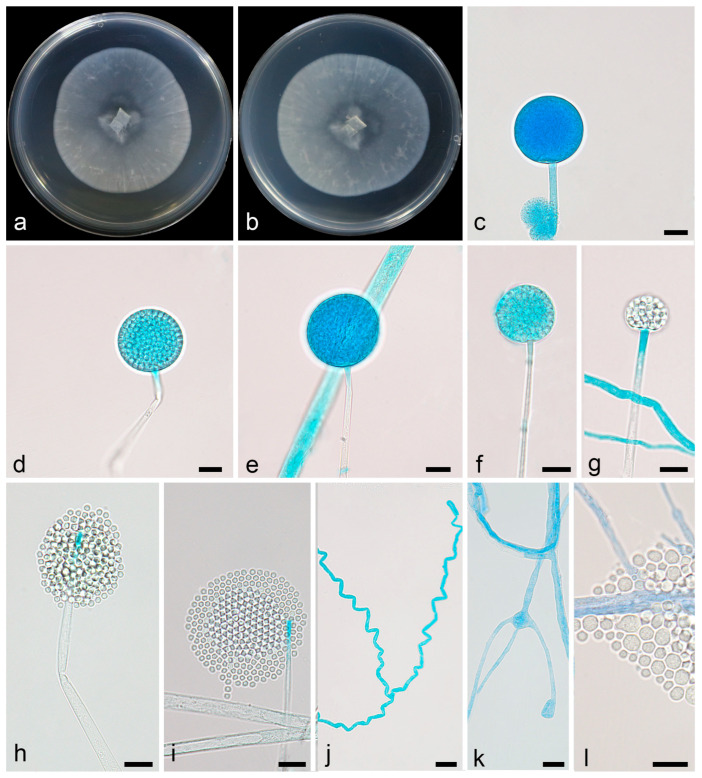
*Mortierella tibetensis* ex-holotype CGMCC 3.28763. (**a**,**b**) Colonies on PDA (**a** obverse, **b** reverse); (**c**–**g**) sporangia; (**h**,**i**) deliquescent sporangia releasing sporangiospores; (**j**) curved hyphae; (**k**) typical swollen hyphae; (**l**) sporangiospores; scale bars: (**c**–**l**) 10 µm.

**Table 1 microorganisms-13-01330-t001:** Data on sequencing the genomic regions of fungal strains.

Loci	PCR Primers	Primer Sequence (5′–3′)	PCR Cycles	References
ITS	ITS5	GGA AGT AAA AGT CGT AAC AAG G	95 °C 5 min; (95 °C: 30 s, 55 °C: 30 s, 72 °C: 1 min) × 35 cycles; 72 °C 10 min	[32]
	ITS4	TCC TCC GCT TAT TGA TAT GC		
LSU	LR0R	GTA CCC GCT GAA CTT AAG C	95 °C 5 min; (94 °C: 30 s, 52 °C: 45 s, 72 °C: 1.5 min) × 30 cycles; 72 °C 10 min	[33]
	LR5	TCC TGA GGG AAA CTT CG		
*RPB1*	RPB1-Af	GAR TGY CCD GGD CAY TTY GG	95 °C 3 min; (94 °C: 40 s, 60 °C: 40 s, 72 °C: 2 min) × 9 (94 °C: 45 s, 55 °C: 1.5 min, 72 °C: 2 min) × 37 cycles; 72 °C 10 min	[34]
	RPB1-Cr	CCN GCD ATN TCR TTR TCC ATR TA		
*Act*	ACT-1	TGG GAC GAT ATG GAI AAI ATC TGG CA	95 °C 3 min; (95 °C: 60 s, 55 °C: 60 s, 72 °C: 1 min) × 30 cycles; 72 °C 10 min	[35]
	ACT-4R	TC ITC GTA TIC TIG CTI IGA IAT CCA CA T		
SSU	NS1	GTA GTC ATA TGC TTG TCT CC	95 °C, 5 min; (94 °C: 60 s, 54 °C: 50 s, 72 °C: 1 min) × 37 cycles; 72 °C 10 min	[32]
	NS4	CTT CCG TCA ATT CCT TTA AG		

## Data Availability

The sequences of this study have been submitted to the NCBI database (https://www.ncbi.nlm.nih.gov/, accessed 15 February 2025), with accession numbers shown in Appendix A.

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
