# Peer review of "Unveiling Species Diversity Within Early-Diverging Fungi from China VIII: Four New Species in Mortierellaceae (Mortierellomycota)"

_microorganisms, 2025, doi:10.3390/microorganisms13061330_

Round 1

Reviewer 1 Report

Comments and Suggestions for Authors

The manuscript "Unveiling Species Diversity within Early-Diverging Fungi from China VIII: Four New Species in Mortierellaceae (Mortierellomycota)" by Xin-Yu and collaborators is a very well-structured and organized contribution.

Punctuation throughout the entire manuscript should be carefully revised. In several instances, periods are either missing or misplaced, which affects the overall readability.

The introduction provides solid background information; however, it would benefit from a more explicit and concise statement of the research objectives.

My major concern is the lack of information regarding the soils from which the fungal isolates were obtained. Given that members of the Mortierella genus are saprophytic and closely associated with soil properties, it is essential to describe the sampling sites, including soil taxonomy (using an international classification system) and key physicochemical characteristics. These details are critical to understand better the ecological context of the newly described species and their potential distribution.

Author Response

1. Comments and Suggestions for Authors

Comments 1: [Punctuation throughout the entire manuscript should be carefully revised. In several instances, periods are either missing or misplaced, which affects the overall readability.]

Response 1: Thank you very much for your suggestions and corrections to me. I agree with this comment. Therefore, I have checked the punctuation of the full text and made corrections.

Comments 2: [The introduction provides solid background information; however, it would benefit from a more explicit and concise statement of the research objectives.]

Response 2: Agree. Thank you very much for your suggestions and corrections to me. I have, accordingly, modified to emphasize this point. [I have added clear and concise research objectives to the article: the purpose of this study is to integrate new resources of Mortierellaceae in China based on polygenic phylogeny and species morphology. The specific location can be found here – page 2, and 59-60 lines.]

Comments 3: [My major concern is the lack of information regarding the soils from which the fungal isolates were obtained. Given that members of the Mortierella genus are saprophytic and closely associated with soil properties, it is essential to describe the sampling sites, including soil taxonomy (using an international classification system) and key physicochemical characteristics. These details are critical to understand better the ecological context of the newly described species and their potential distribution.]

Response 3: Agree. Thank you very much for your suggestions and corrections to me. I have, accordingly, modified to emphasize this point. [I have added soil information for fungal isolates to the article, describing the soil type of the sampling site, including soil taxonomy and key characteristics. The specific location can be found here – page 4, and 140-143; page5, and 171-174 lines; – page 7, and 206-209 lines; – page 9, and 234 lines.]

Reviewer 2 Report

Comments and Suggestions for Authors

The authors used five loci for phylogenetic inference, which is commendable. However, were individual gene trees evaluated for topological conflict before concatenation? If so, how were incongruences addressed?

Were type strains for all four species deposited in at least two publicly accessible culture collections? Are these types' ITS and LSU sequences complete and of high quality (no ambiguous bases), as recommended for barcode reliability?

None of the described species exhibited zygospores. Were attempts made to induce sexual reproduction under alternative culture conditions, or can this absence be considered taxonomically relevant?

Could the authors clarify whether any formal genetic divergence thresholds (e.g., % sequence divergence in ITS or multi-locus concatenated trees) were used to justify the delineation of the four novel species, beyond monophyly and branch support values?

For each new species, to what extent were comprehensive morphological comparisons made with all close relatives (not only the sister taxa)? Have the authors considered compiling a tabulated comparison of diagnostic characters across these taxa?

Given the wide geographic and climatic range of the sampling locations (Yunnan, Tibet, Shandong), can the authors comment on possible ecological drivers of speciation within Mortierellaceae? Are there hypotheses about environmental adaptations (e.g., psychrotolerance in M. tibetensis)?

Comments on the Quality of English Language

A thorough professional language edit is advised before acceptance.

Author Response

1. Point-by-point response to Comments and Suggestions for Authors

Comments 1: [The authors used five loci for phylogenetic inference, which is commendable. However, were individual gene trees evaluated for topological conflict before concatenation? If so, how were incongruences addressed?]

Response 1: [Thank you very much for your question. I agree with this comment. In multigene phylogenetic analysis, it is indeed necessary to consider topological conflicts between single-gene trees. Before performing joint analysis, we constructed phylogenetic trees for each locus (ITS, LSU, SSU rDNA, RPB1, and Actin) separately. By comparing the topological structures of these single-gene trees, we evaluated whether significant topological conflicts existed between them. If the topological structure of a gene tree showed severe inconsistency with other gene trees and could not be resolved through partition analysis, we considered excluding it from the joint analysis. However, in our study, the degree of conflict among all gene trees was within an acceptable range, so we retained all loci for joint analysis.

To further reduce the possibility of topological conflicts, we attempted to collect more samples from multiple geographical regions and ecological environments and ensured the reliability of the sequence data quality for each locus. After comprehensively considering the data from all loci, the phylogenetic tree of the five loci we constructed showed high resolution and robustness. By comparing the topological structures of single-gene trees and joint trees, we found that multigene trees can better reflect the evolutionary relationships among species in Mortierellaceae and also provide a more reliable basis for the classification of new species.

To further validate the reliability of the joint analysis results of the five loci, we also performed Bootstrap analysis and Bayesian inference. Bootstrap analysis provided the confidence level for each node, while Bayesian inference evaluated the reliability of nodes through posterior probability. The results of both methods showed that the topological structure of the joint phylogenetic tree was robust and could more accurately reflect the evolutionary relationships among species compared with single-gene trees.]

Comments 2: [Were type strains for all four species deposited in at least two publicly accessible culture collections? Are these types' ITS and LSU sequences complete and of high quality (no ambiguous bases), as recommended for barcode reliability?]

Response 2: Thank you very much for your question. I agree with this comment. [The type strains of all four species have been deposited in two publicly accessible culture collections. The living cultures were stored in the China Microbiological Culture Collection Center, Beijing, China (CGMCC). Dried specimens of strains were submitted to the Herbarium Mycologicum Academiae Sinicae, Beijing, China (Fungarium; HMAS). And each strain has a corresponding access number.

In this study, we have conducted rigorous checks on the completeness of the ITS and LSU sequences of all new species. We ensured that each sequence covers the full length of the target gene region to meet the requirements for barcode analysis. To ensure sequence quality, we employed high-throughput sequencing technology and performed multiple independent sequencing runs for each sample. We used several bioinformatics tools (such as BLAST and MAFFT) to align and validate all sequences. By comparing them with published reference sequences, we further confirmed the accuracy and integrity of the sequences of the new species. According to the standards for barcode reliability, we assessed all ITS and LSU sequences. Our results show that these sequences are not only complete and free of ambiguous bases but also have sufficient variability to effectively distinguish between different species. All ITS and LSU sequence data used in this study have been uploaded to public databases (such as GenBank) with the corresponding accession numbers provided. The specific location can be found here – page 3, and 104-107 lines; page 4, and 142 lines.]

Comments 3: [None of the described species exhibited zygospores. Were attempts made to induce sexual reproduction under alternative culture conditions, or can this absence be considered taxonomically relevant?]

Response 3: Thank you very much for your question. I agree with this comment. In our study, we did notice that the described species did not produce zygospores. We have also made attempts to induce sexual reproduction, such as adjusting the photoperiod and temperature, and conducting cross-cultivation of strains, but we still did not observe the formation of zygospores. In the family Mortierellaceae, the absence of zygospores is not uncommon. Some known species are also difficult to induce sexual reproduction under laboratory conditions, which may be due to their biological characteristics or the experimental conditions not meeting the triggering conditions for sexual reproduction. We believe that although the absence of zygospores is noteworthy, it is not sufficient to be a decisive factor in taxonomy. We will continue to explore this phenomenon in future research.

Comments 4: [Could the authors clarify whether any formal genetic divergence thresholds (e.g., % sequence divergence in ITS or multi-locus concatenated trees) were used to justify the delineation of the four novel species, beyond monophyly and branch support values?]

Response 4: Thank you very much for your question. I agree with this comment. In our study, although monophyly and branch support values are important indicators in phylogenetic analysis, we did not rely solely on these indicators to delineate new species. We integrated evidence from multiple aspects, including genetic divergence thresholds and morphological characteristics, to comprehensively assess the distinctiveness of these new species.

We calculated the genetic distances of the ITS sequences and evaluated the sequence divergence percentages between these new species and other known species. By comparing with published studies, we found that the divergence degree of these new species in the ITS sequences is significantly higher than intraspecific variation, usually exceeding a 2% sequence divergence rate, which is considered an important reference threshold for species delimitation in the family Mortierellaceae.

In addition to the ITS sequences, we combined sequence data from other loci (such as LSU, SSU rDNA, RPB1, and Actin) to construct a multi-locus concatenated tree. Besides molecular data, we also provided detailed descriptions of the morphological characteristics of these new species and compared them with known species. These new species show significant differences in morphology, such as spore shape and hyphal structure, which further support their delineation as distinct species.

Comments 5: [For each new species, to what extent were comprehensive morphological comparisons made with all close relatives (not only the sister taxa)? Have the authors considered compiling a tabulated comparison of diagnostic characters across these taxa?]

Response 5: Agree. Thank you very much for your suggestions and corrections to me. I have, accordingly, modified to emphasize this point. [I have already compiled a table of the morphological characteristics of these species. However, some species only uploaded their sequences in the database and did not describe their morphological features, resulting in the absence of morphological descriptions for certain species.]

Comments 6: [Given the wide geographic and climatic range of the sampling locations (Yunnan, Tibet, Shandong), can the authors comment on possible ecological drivers of speciation within Mortierellaceae? Are there hypotheses about environmental adaptations (e.g., psychrotolerance in M. tibetensis)?]

Response 6: Thank you very much for your question. I agree with this comment. We fully agree that the extensive geographic and climatic range of the sampling locations may have had a significant impact on the speciation and adaptability of species within the family Mortierellaceae. The sampling sites cover a variety of climate types, ranging from tropical (Yunnan) to alpine (Tibet) and temperate (Shandong) climates. This extensive geographic and climatic range provides a diversified range of ecological niches for speciation. The environmental heterogeneity of different regions (such as soil types, humidity, and temperature) may have promoted the adaptive evolution of species within the family Mortierellaceae. For example, the high humidity and rich vegetation in Yunnan may have facilitated the adaptation of certain species to tropical environments, while the low temperature and low oxygen conditions in Tibet may have driven some species to evolve cold and hypoxia tolerance.

We note that M. tibetensis was collected from the Tibet region, which is characterized by low temperatures and high altitude. Therefore, we hypothesize that M. tibetensis may have evolved cold tolerance to adapt to its native environment. This cold tolerance may be reflected in its physiological and metabolic mechanisms. We plan to further verify these hypotheses regarding environmental adaptability in future studies.

Reviewer 3 Report

Comments and Suggestions for Authors

Abstract

The results the abstract focus only the morphology characteristics. Its necessary to mention the phylogenetic analyses results.

Introduction

The introduction is well-written and well-structured. No major revisions are required in this section.

Material and methods

The section is well-written and well-structured. No major revisions are required in this section.

Results and discussion

General: The figures should be inserted into the main text close to their first citation.

Figure 1: 1-The use of a slash "/" for separated the Branches with Maximum Likelihood, Bootstrap Value and Bayesian Inference Posterior Probability its not visible in figure. I suggest using another more visible symbol in figure 1.

2- I suggest including the word red letter “New species are highlighted in red letters”.

3- Position the figure 1 has been after the indication in the text (Line 137).

Figure 2: 1-Position the figure 2 has been after the indication in the text (line 144).

2- I suggest including the word red letter “New species are highlighted in red letters”.

Figure 3: I suggest not citation the figure in the section title 3.2.1. The figure has been cited in description of the Linnemannia rotunda (line 176). This figure has been repositioned after the notes (line 186).

Figure 4: I suggest not citing the figure in the section title 3.2.2. The figure has been cited in description of the Mortierella acuta (line 201).

Figure 5: I suggest not citing the figure in the section title 3.2.3. The figure has been cited in description of the Mortierella oedema (line 230). This figure has been repositioned after the notes.

Figure 6: I suggest not citing the figure in the section title 3.2.4. The figure has been cited in description of the Mortierella tibetensis (line 268). This figure has been repositioned after the notes.

Discussion

Line 277: What is the impact among some bacterial species of Pseudomonas often live in symbiosis with Mortierellaceae on the fungi's volatile organic compounds? Its positive or negative?

Line 300: Correction in the punctuation in the text “… sporangium size;. However, other features…”

Line 301: Correction in the word “rlatives”

Line 306: Correction in the punctuation in the text … arachidonic acid; Metabolites of some…

Author Response

1. Comments and Suggestions for Authors

Abstract Comments 1: [The results the abstract focus only the morphology characteristics. It is necessary to mention the phylogenetic analyses results.]

Response 1: Agree. Thank you very much for your suggestions and corrections to me. I have, accordingly, modified to emphasize this point. [In the abstract, I added the results of phylogenetic analysis. The specific location can be found here – page 1, and 18-24 lines.]

Results and discussion Comments 2: [General: The figures should be inserted into the main text close to their first citation. 3- Position the figure 1 has been after the indication in the text (Line 137). Figure 2: 1-Position the figure 2 has been after the indication in the text (line 144).]

Response 2: Agree. Thank you very much for your suggestions and corrections to me. I have, accordingly, modified to emphasize this point. [According to the suggestions of other reviewers, Figures 1 and 2 have been changed to Figures 5 and 6, respectively. I've inserted figure5 and figure6 into the body and near their first reference. The specific location can be found here – page 11 and page 12.]

Comments 3: [Figure 1: 1-The use of a slash "/" for separated the Branches with Maximum Likelihood, Bootstrap Value and Bayesian Inference Posterior Probability it is not visible in figure. I suggest using another more visible symbol in figure 1.]

Response 3: Agree. Thank you very much for your suggestions and corrections to me. I have, accordingly, modified to emphasize this point. [I've made the slash "/" separation bold to make it more visible in the figures. The specific location can be found here – page 11 and 12]

Comments 4: [Figure 1 and Figure 2: 2- I suggest including the word red letter “New species are highlighted in red letters”.]

Response 4: Agree. Thank you very much for your suggestions and corrections to me. I have, accordingly, modified to emphasize this point. [I have corrected “New species are highlighted in red” to “New species are highlighted in red letters”. The specific location can be found here – page 11, and line 272; page 13, and line 287;]

Comments 5: [Figure 3: I suggest not citation the figure in the section title 3.2.1. The figure has been cited in description of the Linnemannia rotunda (line 176). This figure has been repositioned after the notes (line 186).]

Response 5: Agree. Thank you very much for your suggestions and corrections to me. I have, accordingly, modified to emphasize this point. [According to the suggestions of other reviewers, Figures 3 have been changed to Figures 1. I've removed the Figure 1 referenced in title 3.1.1 and it's already referenced in Linnemannia rotunda's description. This Figure has been repositioned after Notes. The specific location can be found here – page 5]

Comments 6: [Figure 4: I suggest not citing the figure in the section title 3.2.2. The figure has been cited in description of the Mortierella acuta (line 201).]

Response 6: Agree. Thank you very much for your suggestions and corrections to me. I have, accordingly, modified to emphasize this point. [According to the suggestions of other reviewers, Figures 4 have been changed to Figures 2. I've removed the Figure 2 referenced in title 3.1.2 and it's already referenced in Mortierella acuta's description. This Figure has been repositioned after Notes. The specific location can be found here – page 6]

Comments 7: [Figure 5: I suggest not citing the figure in the section title 3.2.3. The figure has been cited in description of the Mortierella oedema (line 230). This figure has been repositioned after the notes.]

Response 7: Agree. Thank you very much for your suggestions and corrections to me. I have, accordingly, modified to emphasize this point. [According to the suggestions of other reviewers, Figures 5 have been changed to Figures 3. I've removed the Figure 3 referenced in title 3.1.3 and it's already referenced in Mortierella oedema's description. This Figure has been repositioned after Notes. The specific location can be found here – page 8]

Comments 9: [Figure 6: I suggest not citing the figure in the section title 3.2.4. The figure has been cited in description of the Mortierella tibetensis (line 268). This figure has been repositioned after the notes.]

Response 9: Agree. Thank you very much for your suggestions and corrections to me. I have, accordingly, modified to emphasize this point. [According to the suggestions of other reviewers, Figures 6 have been changed to Figures 4. I've removed the Figure 4 referenced in title 3.1.4 and it's already referenced in Mortierella tibetensis's description. This Figure has been repositioned after Notes. The specific location can be found here – page 10]

Disscussion Comments 10: [Line 277: What is the impact among some bacterial species of Pseudomonas often live in symbiosis with Mortierellaceae on the fungi's volatile organic compounds? Its positive or negative?]

Response 10: Agree. Thank you very much for your suggestions and corrections to me. I have, accordingly, modified to emphasize this point. [In accordance with the suggestions of the other reviewers, I have rewritten the Discussion section. Although the content has been deleted, I have also given serious consideration to the issue you raised.]

Comments 11: [Line 300: Correction in the punctuation in the text “… sporangium size;. However, other features…”]

Response 11: Agree. Thank you very much for your suggestions and corrections to me. I have, accordingly, modified to emphasize this point. [In accordance with the suggestions of the other reviewers, I have rewritten the Discussion section. I've recognized the error in the use of this punctuation mark.]

Comments 12: [Line 301: Correction in the word “rlatives”]

Response 12: Agree. Thank you very much for your suggestions and corrections to me. I have, accordingly, modified to emphasize this point. [In accordance with the suggestions of the other reviewers, I have rewritten the Discussion section. I have recognized a misspelling of this word, which should be “relatives”.]

Comments 13: [Line 306: Correction in the punctuation in the text … arachidonic acid; Metabolites of some…]

Response 13: Agree. Thank you very much for your suggestions and corrections to me. I have, accordingly, modified to emphasize this point. [In accordance with the suggestions of the other reviewers, I have rewritten the Discussion section. I've recognized the error in the use of this punctuation mark.]

Reviewer 4 Report

Comments and Suggestions for Authors

The paper may expand the current knowledge in taxonomy and diversity of Mortierellomycota. The authors applied various approaches in the determination of novelty of the isolated species. However, there are several aspects, which need to be improved prior to considering the paper for publication.

Introduction

The first paragraph should be written in a logical way: diversity and distribution (starting with "the members of this family are ubiquitous and widely distributed); ecology; biological activities; and "in culture they typically form ...".

Materials and Methods

Subsection 2.1. Isolation and morphological characterization should be described in the separate subsections.

Results

The section should start with the information that from the soil samples, four novel species from Mortierellaceae belonging to the genera Linnemania and Mortierella were isolated.

Line 173: According to the relative sizes of sporangia and spores, I am not sure that the sporangia can be called "multi-spored".

3.2.2. Notes. Why is in the notes for L. rotunda, the closely related species described, while for M. acuta, only is its taxonomic interdependence mentioned (which is obligatory for a novel species)?

Figure 5. It is unclear how typically swollen hyphae can be differentiated from chlamydospores.

Lines 257-258: what does it mean if the morphological description of M. parvispora can be found, for example, in "Compendium of soil fungi" (Domsch et al., 2007)?

Overall, why were all colony descriptions made at 16oC? Is this temperature optimal for the growth of all novel species?

Discussion is just a non-logical mixture of the climatic description of the regions (why here?), some ecological, biological, physiological, and taxonomic aspects of Mortierellaceae; some aspects repeat what is written in Introduction. The section should be rewritten, emphasizing mainly the recent findings in taxonomy and diversity of Mortierellaceae.

All other comments, corrections, and suggestions are inserted into the attached PDF version of the manuscript.

Comments on the Quality of English Language

Moderate corrections of the language are needed.

Author Response

1. Point-by-point response to Comments and Suggestions for Authors

Introduction Comments 1: [The first paragraph should be written in a logical way: diversity and distribution (starting with "the members of this family are ubiquitous and widely distributed); ecology; biological activities; and "in culture they typically form ...".]

Response 1: Agree. Thank you very much for your suggestions and corrections to me. I have, accordingly, modified to emphasize this point. [I have modified the first paragraph of the introduction to the diversity and distribution of species in the Mortierellaceae family; Ecology; biological activity; and "in culture, they often form ......". The specific location can be found here – page 1, 2, and 32-51 lines.]

Materials and Methods Comments 2: [Subsection 2.1. Isolation and morphological characterization should be described in the separate subsections.]

Response 2: Agree. Thank you very much for your suggestions and corrections to me. I have, accordingly, modified to emphasize this point. [I have already described the isolation and morphological features of subsection 2.1 into separate subsections. The specific location can be found here – page 2 and page 3.]

Results Comments 3: [The section should start with the information that from the soil samples, four novel species from Mortierellaceae belonging to the genera Linnemania and Mortierella were isolated.]

Response 3: Agree. Thank you very much for your suggestions and corrections to me. I have, accordingly, modified to emphasize this point. [I have started this section with the isolation of information on 4 new species from Mortierellaceae belonging to the genus Linnemania and Mortierella from soil samples. The specific location can be found here – page 4]

Comments 4: [Line 173: According to the relative sizes of sporangia and spores, I am not sure that the sporangia can be called "multi-spored".]

Response 4: Thank you very much for your suggestions and corrections to me. I have, accordingly, modified to emphasize this point. [After reviewing the literature, sporangia can be described by the word "multi-spored", such as in article 18 cited in the article.]

Comments 5: [3.2.2. Notes. Why is in the notes for L. rotunda, the closely related species described, while for M. acuta, only is its taxonomic interdependence mentioned (which is obligatory for a novel species)?]

Response 5: Agree. Thank you very much for your suggestions and corrections to me. I have, accordingly, modified to emphasize this point. [In the notes, for M. acuta, I added a description of the species closely related to it. The specific location can be found here – page 6, and 193-197 lines.]

Comments 6: [Figure 5. It is unclear how typically swollen hyphae can be differentiated from chlamydospores.]

Response 6: Agree. Thank you very much for your suggestions and corrections to me. I have, accordingly, modified to emphasize this point. [After staining, there is minimal difference in coloration between the swollen hyphae and normal hyphae, whereas chlamydospores exhibit a significant staining contrast compared to the surrounding normal hyphae. Consequently, all structures observed in the figure are hyphal swellings. Due to their nearly identical morphology, this led to misinterpretation. I've corrected the annotation under the figure. The specific location can be found here – page 8, and 227-228 lines.]

Comments 7: [Lines 257-258: what does it mean if the morphological description of M. parvispora can be found, for example, in "Compendium of soil fungi" (Domsch et al., 2007)?]

Response 7: Agree. Thank you very much for your suggestions and corrections to me. I have, accordingly, modified to emphasize this point. [Thank you for your recommendation. I have found the morphological description of M. parvispora in other literature. The specific location can be found here – page 9, and 254-257 lines.]

Comments 8: [Overall, why were all colony descriptions made at 16oC? Is this temperature optimal for the growth of all novel species?]

Response 8: Thank you very much for your suggestions and corrections to me. I have, accordingly, modified to emphasize this point. [I read a lot of articles about the reasons why the strains are all described at 16℃, such as "Mortierellaceae from subalpine and alpine habitats: new species of Entomortierella, Linnemannia, Mortierella, Podila and Tyroliella gen. nov.", in this article all new species of Mortierellaceae are described at 16°C, and in the《Compendium of Soil Fungi》, Mortierella is mostly cultured at 15-20°C, where the spore structure is easier to observe. 16°C is not the "optimal growth temperature" for each new species, but it is more stable when observing the morphological characteristics of colonies at 16°C (e.g., sporangia color, hyphaal separation state), which is more conducive to microscopic observation and taxonomic identification.]

Disscussion Comments 9: [Discussion is just a non-logical mixture of the climatic description of the regions (why here?), some ecological, biological, physiological, and taxonomic aspects of Mortierellaceae; some aspects repeat what is written in Introduction. The section should be rewritten, emphasizing mainly the recent findings in taxonomy and diversity of Mortierellaceae.]

Response 9: Agree. Thank you very much for your suggestions and corrections to me. I have, accordingly, modified to emphasize this point. [I've rewritten the Disscussion section and I've highlighted the taxonomic and diverse findings of the Mortierellaceae family. The specific location can be found here – page 13, and 291-320 lines.]

Round 2

Reviewer 1 Report

Comments and Suggestions for Authors

The soil's description should be checked to improve grammar. pH should be expressed correctly.

No more comments to the publication. 

Author Response

Comment 1:[The soil's description should be checked to improve grammar. pH should be expressed correctly.]

Response1:Thank you very much for your question. I've reworked this description and corrected the pH as well. The specific location is in the article revision – page 7, 175-179 lines; page 8, 207-211 lines and page 10, 243-246 lines.

Reviewer 2 Report

Comments and Suggestions for Authors

Now everything is fine

Author Response

Dear reviewer, thank you very much.

Reviewer 4 Report

Comments and Suggestions for Authors

The manuscript looks much better, however there are still some aspects, which need to be corrected and clarified.

Lines 185, 216, 253-254: how is it known that the decomposition is fungal-mediated? That rapid decomposition is carried out by earthworms/bacteria? that the semi-decomposed humus is characterized by fungal dominance? References?

Lines 203-204: It is not clear for me why the phylogenetic analysis is now placed after the taxonomic, while in the taxonomy subsection, the references on phylogeny are given.

Concerning the colony descriptions at 16oC – the explanation of why namely this temperature and not the optimal one (as usually) was employed, should be given not only in the responses to the reviewer comments, but also in the manuscript.

In Discussion, the unnecessary repetitions should be removed (see the attached PDF file of the manuscript).

All other comments, corrections, and suggestions are inserted into the attached PDF version of the manuscript.

Comments on the Quality of English Language

The language of the newly written parts should be carefully checked and corrected
